# Control of capillary behavior through target-responsive hydrogel permeability alteration for sensitive visual quantitative detection

Yansheng Li[1], Yanli Ma[2], Xiangyu Jiao[1], Tingyu Li[2], Zhehao Lv[2], Chaoyong James Yang [2,3], Xueji Zhang[1] & Yongqiang Wen[1]

DNA hydrogels have received considerable attention in analytical science, however, some limitations still exist in the applications of intelligent hydrogels. In this paper, we describe a way to prepare gel film in a capillary tube based on the thermal reversible principle of DNA hydrogel and the principle of capillary action. Because of the slight change in the internal structure of gel, its permeability can be increased by the addition of some specific targets. The capillary behavior is thus changed due to the different permeability of the hydrogel film. The duration time of the target solution flowing through the capillary tube with a specified length is used to quantify this change. With this proposed method, ultra-trace DNA hydrogel (0.01 μL) is sufficient to realize the sensitive detection of cocaine without the aid of other instruments, which has a low detection limit (1.17 nM) and good selectivity.

[1] Research Center for Bioengineering and Sensing Technology, School of Chemistry & Biological Engineering, University of Science and Technology Beijing, Beijing 100083, PR China. [2] MOE Key Laboratory of Spectrochemical Analysis & Instrumentation, Collaborative Innovation Center of Chemistry for Energy Materials, Key Laboratory for Chemical Biology of Fujian Province, State Key Laboratory of Physical Chemistry of Solid Surfaces, College of Chemistry and Chemical Engineering, Xiamen University, Xiamen 361005, PR China. [3] Institute of Molecular Medicine, Renji Hospital, Shanghai Jiao Tong University, School of Medicine, Shanghai 200127, PR China. Correspondence and requests for materials should be addressed to C.J.Y. (email: cyyang@xmu.edu.cn) or to X.Z. (email: zhangxueji@ustb.edu.cn) or to Y.W. (email: wyq_wen@ustb.edu.cn)

DNA hydrogels are based on the interlocking of assembled DNA secondary structures that undergo swelling in water[1], where DNA not only serves as a reversible cross-linker modulating the mechanical and rheological properties of hydrogels but also selectively responds to a variety of different molecules[1–7]. Moreover, DNA hydrogels are convenient to store and can immobilize reactive functional groups, chemicals, or cells without losing their physicochemical property in changeable surrounding environments[8,9]. Over the last decade, aptamer-functionalized DNA hydrogel has elegantly been applied in analysis science due to its excellent portability and molecular recognition properties[10–12]. Recently, DNA hydrogels have been used for the in situ identification of circulating tumor cells by two-color confocal microscopy combined with an aptamer-trigger clamped hybridization chain reaction method[10]. Some DNA hydrogel-based sensors have also been proposed, where the analyte-induced chemical or physical changes of DNA hydrogel are transduced into an electrical or optical signal[13–19]. In addition, in order to reduce the dependence on other analytical instruments, Yang's research group[20–25] designed a series of bar-chart chips for diagnostics based on DNA hydrogel. In their way, they translated DNA hydrogel degradation events into distance-based signals just by monitoring with the naked eye.

However, all those published sensors were designed based on the idea of analyte-dependent formation or complete degradation of DNA hydrogel[12,26,27]. which can usually achieve qualitative detection or need other auxiliary means to help characterize the gel degradation to achieve precise detection. In addition, those studies only focused on macro-morphological changes, without considering micro-changes and processes. It is well known that any change in macroscopical performance comes from the accumulation of microscopic change. The complete degradation of gel into sol is the result of the accumulation of small changes of gel to a certain extent. A small number of target molecules can only cause small changes in DNA hydrogel, but will not cause the degradation of the whole gel. Therefore, the transformation from gel to sol requires more stimulations, resulting in high limit of detection (LOD) in previously published work based on gel degradation. In addition, in order to facilitate the observation and use of sol–gel transformation, gels are usually prepared in large containers and a large volume of DNA hydrogels were used, which would increase the cost of the measurement. These practical shortcomings limit the further application of DNA hydrogels in visual quantitative sensors, especially in rural clinics and patients' homes[9]. How to transform such small changes in DNA hydrogel into readable, visual signals without relying on other equipment is a great challenge to further promote the application of DNA hydrogel.

Herein, we report a simple, cheap, sensitive capillary self-driven regulator sensor (CSDR-Sensor), which can transform the analyte-induced small changes inside the DNA hydrogel into visual signals. As shown in Fig. 1, DNA hydrogel film can be subtly prepared in capillary tube to obtain the CSDR-Sensor based on the thermal reversible principle of DNA hydrogel and the principle of capillary action. In the analysis process, the permeability of the DNA hydrogel film will increase because of the small structural changes in the gel induced by the interaction between target molecules and the aptamer linkers, thereby changing the flow velocity of the sample solution in the capillary tube. The duration time of the target solution flowing through the capillary tube with a specified length is used to characterize the concentration of different solutions. Compared with the previously reported DNA hydrogel sensors, this research has broken the dependence of DNA hydrogel qualitative transformation between gel and sol. By flexibly utilizing capillary tube and capillary action (self-driven motion), this sensor directly converts the target response into an easily measurable parameter by the slight microscopic changes of gel (i.e., permeability alteration) rather than macroscopic shape changes, which greatly reduces the detection limit of analysis. This sensor has also realized the visual quantitative detection without relying on other equipment. Furthermore, it can greatly reduce the usage of gel by using the hydrogel film in capillary tube. The principles and methods used in the preparation and use of CSDR-Sensor have opened a route for the design of visual quantitative sensors and greatly improved the practicability of DNA hydrogel.

## Results

**Design and fabrication of the CSDR-Sensor.** In this work, a cocaine aptamer, which was proposed by Landry's group and had been used to design various biosensors, is used in the design of DNA hydrogel for the detection of model target cocaine in our visual sensing devices[28]. Traditional hydrogel films, as reported in the literature, were normally prepared with a reaction mixture containing monomers and a crosslinking agent, which was either spin-coated onto a planar substrate or confined between two planar substrates[13–15,29]. In this work, the gel film was subtly prepared in capillary tube, which was based on the thermal reversible principle of DNA hydrogel and the principle of capillary action. As shown in Fig. 2, when the pre-prepared DNA

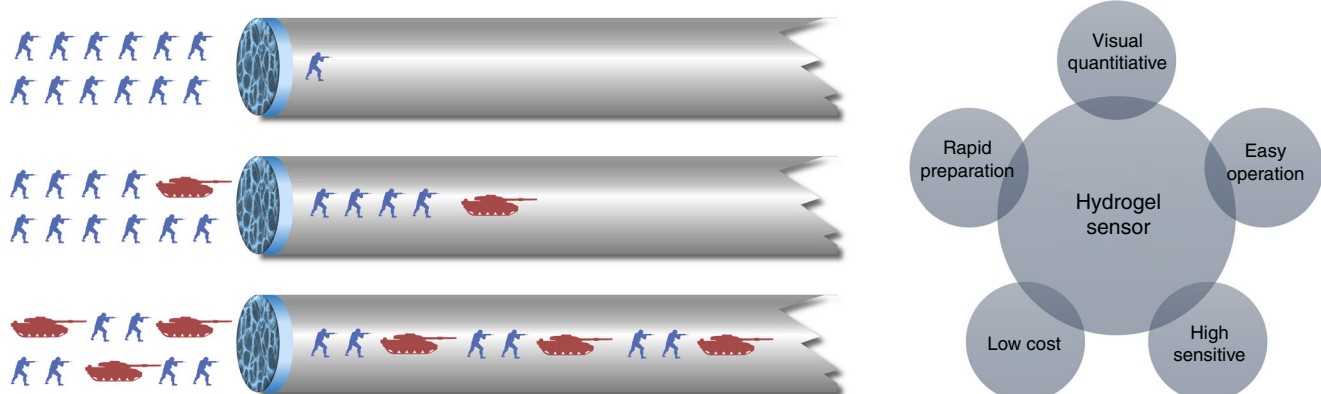

**Fig. 1** The scheme of a target-responsive hydrogel film in capillary tube for visual quantitative detection. The soldiers in the picture represent the solvent molecules that can slowly permeate through the gel membrane, and the tanks in the picture represent the target molecules that can degrade the gel membrane. In the presence of cocaine, cocaine molecules can change the internal structure of DNA hydrogel, resulting in an increase in the permeability of the film and the flow rate of the solvent in the capillary. The higher the concentration of cocaine molecule is, the faster the flow rate is

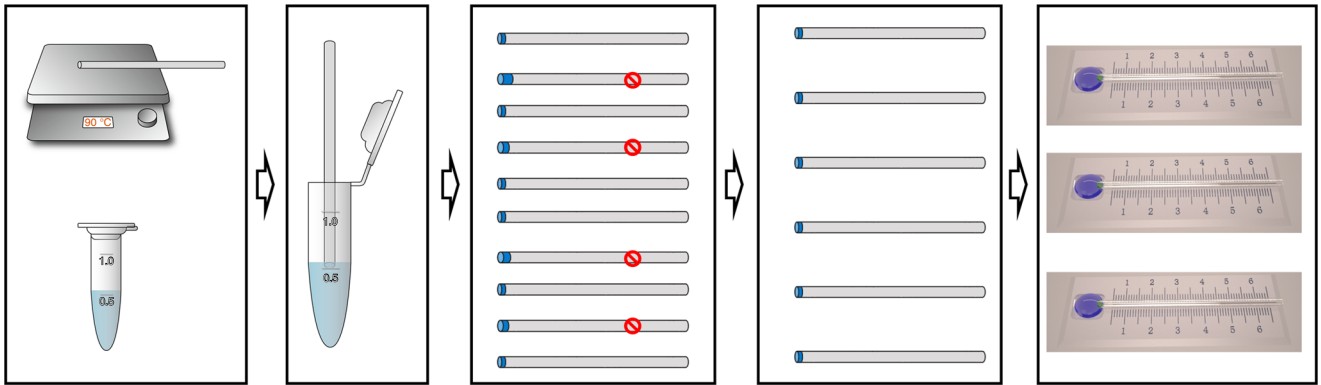

**Fig. 2** The preparation process of the CSDR-Sensor. First, the DNA hydrogel is prepared, and then the capillary tube is placed on the 90 °C heating plate for 10 min. Second, the hot capillary tube is inserted vertically into the prepared DNA hydrogel, and is then lifted up after 3 s. Third, the gel films in capillary tube with a thickness of 0.15 ± 0.01 mm are selected under the microscope. Finally, the capillary tube is fixed on a scale substrate

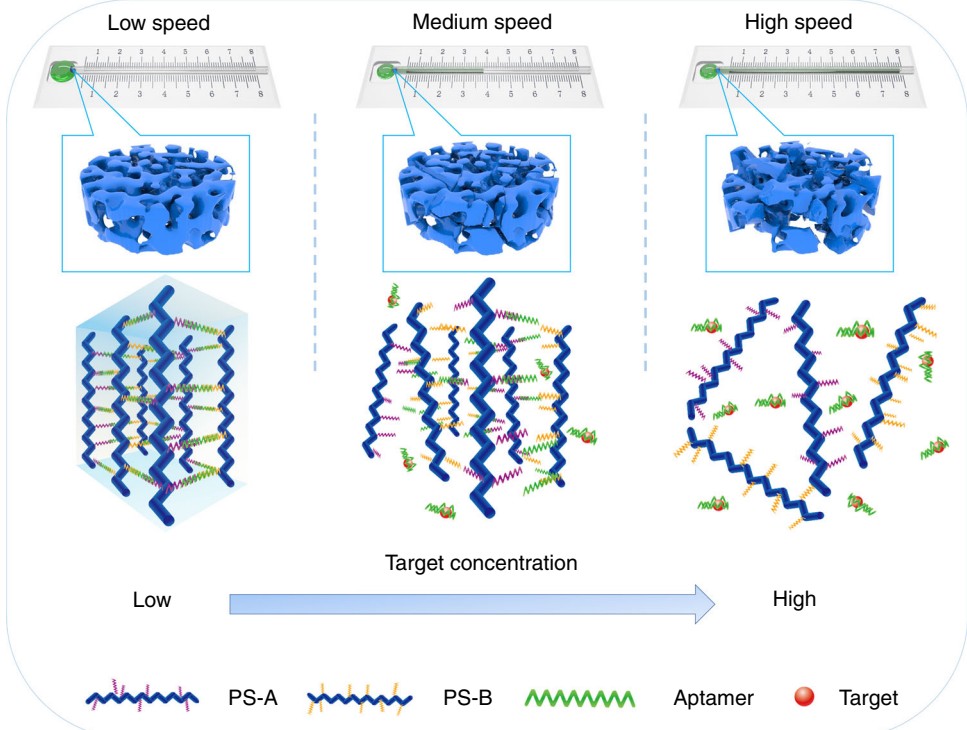

**Fig. 3** The working principle of the CSDR-Sensor. The addition of target will result in the degradation of hydrogel to a certain degree, and then change the permeability of gel in capillary tube and regulate the capillary action via membrane permeability

hydrogel was in contact with the hot capillary tube, it turned into a liquid solution, which was then injected into the capillary tube by virtue of the capillary action. Subsequently, the DNA hydrogel film was formed in the capillary tube at room temperature to obtain the CSDR-Sensor. The gel film of 0.15 ± 0.01 mm was obtained by adjusting the time (3 s) of hot capillaries in the gel. A statistical analysis of the thickness of the gel film was made, and the yield of qualified product was about 90% as shown in Supplementary Figure 2. In order to ensure that the liquid velocity is not affected in our experiments, the error of the thickness of gel film is controlled within ±0.01 mm.

**Working principle of the CSDR-Sensor.** In a typical test using CSDR-Sensor, the device was firstly placed horizontally. The sample solution was added onto a carrier plate, and then moved the carrier plate to touch the hydrogel film in capillary tube. A

timer with a resolution of tens of milliseconds was used to record the flow time required for the sample through the capillary tube with certain length. Hydrogel films are very stable and sensitive and have attracted the attention of increasingly more researchers regarding their potential versatility for the efficient detection or sensing of target molecules[13–16]. However, these DNA hydrogel film sensors are not widely used by the public because they usually rely on expensive large-scale equipment and trained personnel for accurate preparation. Permeability is one of the characteristics of hydrogels that immediately change during gel degradation by external stimuli. The permeability of a gel film is difficult to measure and has not been used for DNA hydrogel sensors because traditional hydrogel films are very soft and easily break[30,31]. In this work, every CSDR-Sensor is composed of a DNA hydrogel film within a capillary tube and a substrate with scale, as shown in Fig. 3. Responsive DNA gels are used to block

the capillary tube and regulate the capillary action via membrane permeability. The capillary tube protects the gel from destruction, and the capillary action allows the permeability of the gel to be visually measured. The cocaine aptamer is designed as a DNA hydrogel linker, which is a DNA molecule that can bind cocaine. DNA sequences used for this work are shown in Supplementary Table 1. In the presence of cocaine, the cocaine aptamer is able to bind the analyte to form the cocaine-stabilized conjugates. Subsequently, the permeability will be increased as a result of reducing the crosslinking density. In contrast, a blank sample (without cocaine) will not produce permeability changes in the DNA hydrogel film. Therefore, the different permeability of the DNA hydrogel results in different flow velocities of the sample solution in the capillary. Thus, the target can be visually and quantitatively detected without an external electronic device and power source. This ingenious method provides a portable, sensitive, and visual quantitative sensor that resolves the faced issues of DNA hydrogel applications.

**Optimization of the aptamer linker.** In this sensing scheme, the pore size of DNA hydrogel network plays a vital role in its sensitivity. By regulating the concentration of DNA aptamer in the hydrogel, the pore size of the hydrogel can be changed, which can optimize the diffusion rate of target molecules in the hydrogel so that the hydrogel can be used for sensitive and rapid detection of targets in a range of concentrations. The pore size of the gel is determined by the concentration of the DNA linker. In order to get the proper gel, optimization experiments were carried out.

The relevant data are shown in Fig. 4a–d. Three well-formed gels prepared from different concentrations of aptamer (50, 70, and 90 μM linker DNA in the hydrogel) were first selected for further optimization to detect 0 and 100 μM cocaine solutions. And the flow-through time over an 8 cm distance for 0 and 100 μM cocaine solutions are recorded as $T(0\,\mu M)$ and $T(100\,\mu M)$ (Fig. 4a, left $y$-axis), respectively. The signal-to-noise ratio (SNR) is defined as the ratio of $\{T(0\,\mu M)-T(100\,\mu M)\}$ to $T(0\,\mu M)$, which distinguish the signal from background levels. As shown in the right $y$-axis in Fig. 4a, the SNRs of DNA crosslinking densities are 0.23 (50 μM linker DNA), 0.93 (70 μM linker DNA), and 0.76 (90 μM linker DNA hydrogel), respectively. Therefore, the DNA hydrogel with 70 μM linker concentration is chosen for the following experiments.

**Characterization of DNA hydrogels.** The properties of the as-prepared hydrogel were then characterized using rheology. In order to obtain the linear viscoelastic region, amplitude scanning (shear strain range 0.01–1000%) was performed for 70 μM DNA hydrogels. Figure 4b shows that the storage modulus and loss modulus are constants when the shear strain range is 0.1–10%, and the characteristic of hydrogels can be obtained within this range. As the DNA hydrogel is based on the interlocking of assembled DNA secondary structures, which are only stable under their melting point, such DNA hydrogel should be temperature dependent. As shown in Fig. 4c, the storage modulus $G'$ value of hydrogel decreases with the increase of temperature and the gelling-transition point is about 40 °C at 70 μM aptamer

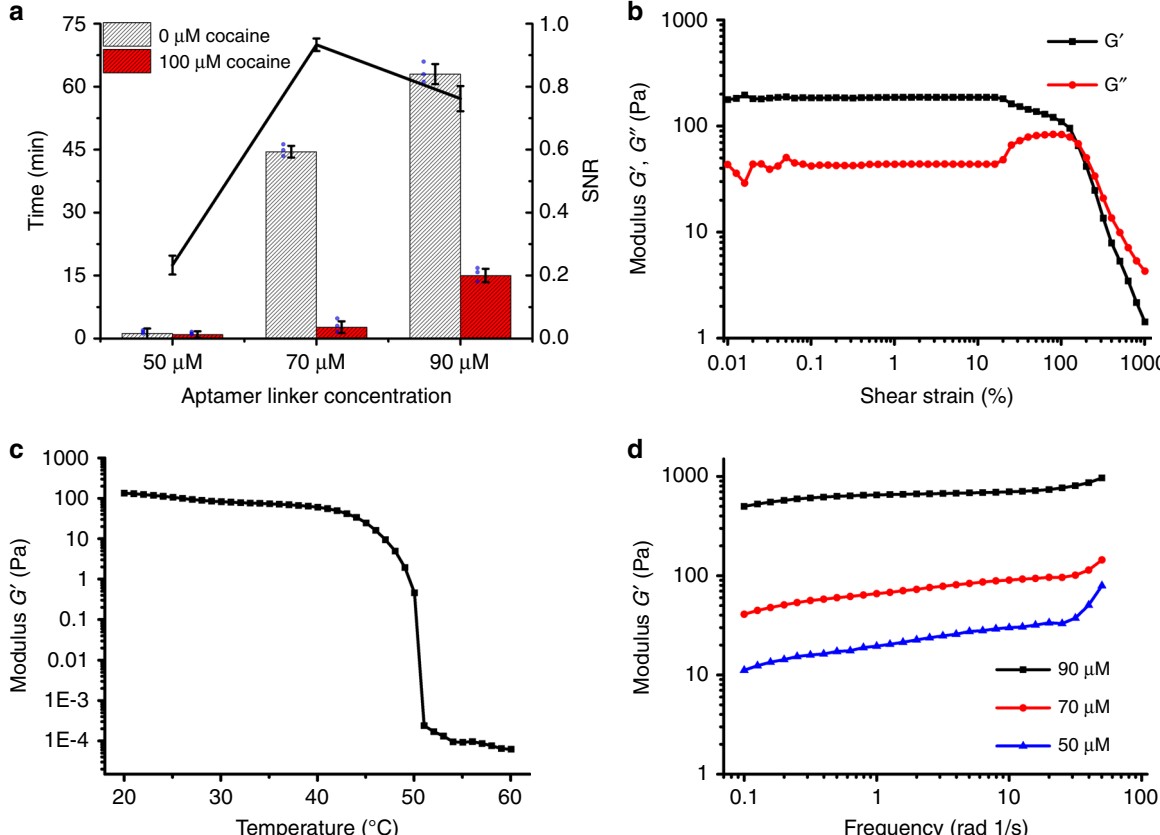

**Fig. 4** Optimization and characterization of DNA hydrogels. **a** The flow-through time (left $y$-axis) and SNRs (right $y$-axis) obtained from different linker concentration of hydrogels for the detection of 0 and 100 μM cocaine using the CSDR-Sensor. Each data point is an average of three replicates ($N = 3$), and the error bars indicate the standard deviations. **b** The storage modulus $G'$ and loss modulus $G''$ of the DNA hydrogel (70 μM aptamer linker) changes with shear strain. **c** Variation of the storage modulus $G'$ value of the hydrogel function of temperature at a linker DNA concentration of 70 μM. **d** Storage modulus $G'$ at 25 °C as a function of shear frequency for DNA hydrogels prepared with different aptamer linker concentration

linker concentration. Frequency sweep tests were performed between 0.1 and 50 rad s$^{-1}$ with a fixed strain of 1% at 25 °C (Fig. 4d). In the frequency sweep, the storage modulus $G'$ value increased from 10 to 900 Pa with the increase of DNA linker concentration, which can be attributed to the densification of the hydrogel network[32,33]. Considering the result of Fig. 4a, the capillary flow velocity of the sample gradually decreases with the increase of aptamer concentration, which is consistent with the storage modulus analysis results. The larger the storage modulus is, the smaller the hydrogel mesh is, and the corresponding permeability becomes lower. The relationship among aptamer concentration, gel permeability and the flow time of samples confirmed the feasibility of controlling capillary flow rate by regulating the permeability of gel membrane, which is the theoretical basis of our experiments.

Subsequently, as shown in Supplementary Figure 1a, the volume of gel (thickness is 0.15 mm) is calculated to be 0.0106 μL based on the inner diameter (0.3 mm) of the capillary. Additionally, the DNA hydrogel shows well-defined interconnected 3D porous networks. From the highly magnified scanning electron microscopy (SEM) images in Supplementary Figure 1b, the network exhibits varying micrometer-sized pores, which suggests that the hydrogel can provide space for the reaction.

**Visual quantitative detection of cocaine**. To further investigate the accuracy of this method based on the aptamer-functionalized DNA hydrogel film, different concentrations of cocaine are detected. Different concentrations of cocaine are spiked in a buffer and detected by the CSDR-Sensor based on the readout time. The cocaine solutions are placed in contact with the gel film in the capillary, and the flow-through time of the sample solution is recorded. The analysis results of the CSDR-Sensor for the detection of cocaine are shown in Fig. 5a–c. Cocaine concentrations ranging from 0 to 100 μM are tested with three repeated measurements, each in parallel. As shown in Fig. 5a, when the 100 μM cocaine sample is assayed, it takes 2′45″ for the sample solution to flow through a 6 cm length. The short flow time indirectly reflects an increasing permeability as a result of the reduction in the crosslinking density induced by high concentrations of the cocaine. On the other hand, for the 10 nM cocaine sample, it takes 15′30″ to allow the solution to flow through a 2.6 cm length, and the solution still does not reach the 6 cm position after 23′30″ (Fig. 5a). As shown in Fig. 5b and in the Movies (Supplementary Movie 1 (0 nM), Supplementary Movie 2 (1 μM). and Supplementary Movie 3 (100 μM) cocaine detection, respectively), the flow rate of the sample in the capillary gradually increases as the cocaine concentration increases. A high concentration of cocaine results in more cocaine molecules binding with the linker in the DNA hydrogel, which leads to an increasing permeability of the DNA hydrogel. Figure 5c shows that the flow-through time is directly proportional to the cocaine concentration. Therefore, the CSDR-Sensor has the excellent capability of quantitative cocaine detection in a linear range from 10 nM to 100 μM. The linear regression equation is obtained as $y = -9.9692x + 51.0845$ with a correlation coefficient of 0.9879. The LOD is calculated to be 1.17 nM based on the 3σ/slope rule ($N = 3$), where σ is the standard deviation of the blank samples. These results clearly demonstrate that the CSDR-Sensor based on the DNA hydrogel film in a capillary tube can achieve quantitative detection with the naked eye with excellent accuracy.

**Evaluation of selectivity and practicality**. The potential false-positive signals will affect the practical performance of the sensor, which should be excluded. Benzoylecgonine (BE) and ecgonine

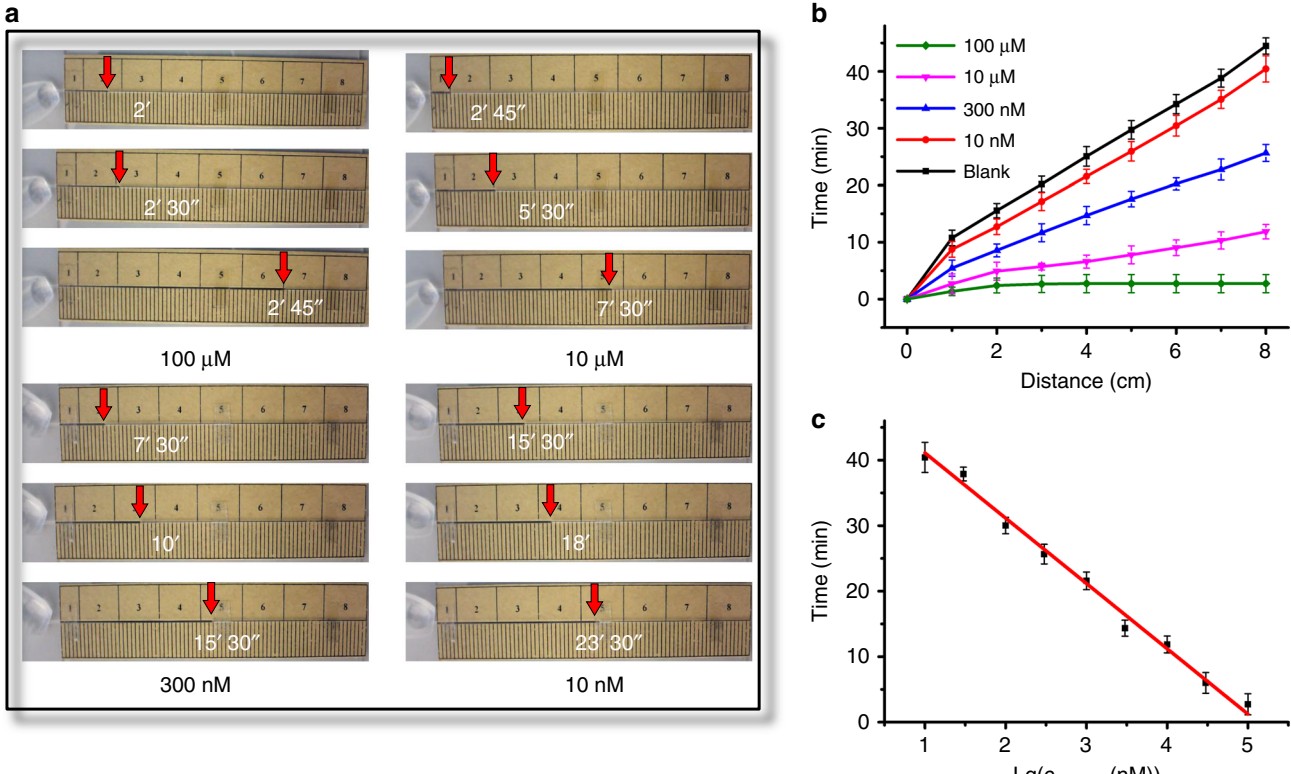

**Fig. 5** Performance of the CSDR-Sensor for the detection of cocaine. **a** Images showing sample solution advancement for the detection of cocaine at different time. **b** Time-dependent sample solution advancement with different concentrations of cocaine. **c** Timing results for the detection of various concentrations of cocaine. Each data point is an average of three replicates ($N = 3$), and the error bars indicate the standard deviations

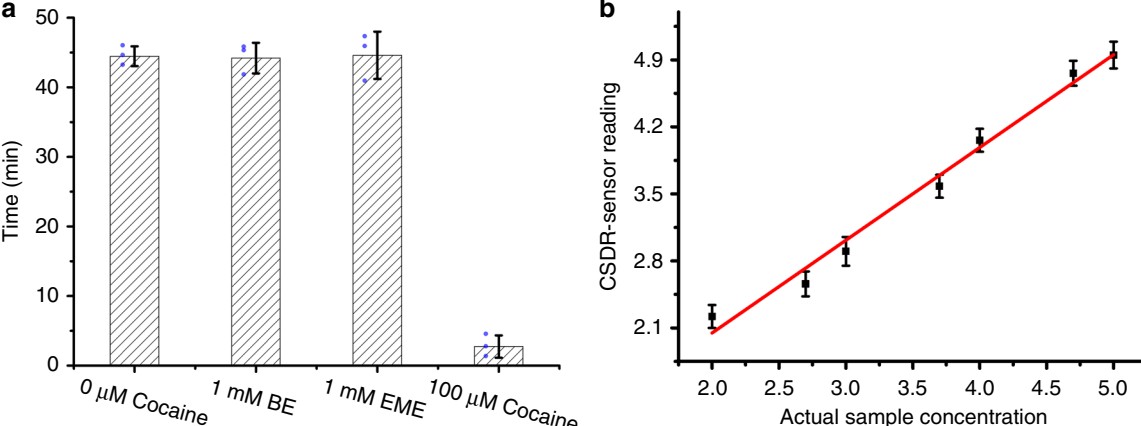

**Fig. 6** Selectivity and practicality of the CSDR-Sensor. **a** Selectivity of the CSDR system with cocaine, benzoylecgonine (BE), and ecgonine methyl ester (EME). **b** Linear standard curves of cocaine detection in urine are obtained from 10 nM to 100 µM in an 8 cm distance in a capillary tube. Each data point is an average of three replicates ($N = 3$), and the error bars indicate the standard deviations

methyl ester (EME) are the most common interfering molecules in the detection of cocaine. In order to test the anti-interference performance of the device in a complex environment, two sample solutions (containing 1 mM BE and 1 mM EME) and a blank sample were tested with our CSDR-Sensors, respectively. The results obtained were compared with those of samples containing 100 µM cocaine. As shown in Fig. 6a, the analytical results of the solution containing BE and EME are basically consistent with those of blank samples. DNA adapter molecules in hydrogels can only specifically recognize cocaine, but cannot recognize BE and EME, even at high BE and EME concentrations. Therefore, the sensor based on DNA hydrogel membrane has good ability to recognize target molecule cocaine in real detection scenarios, and the interfering molecules cannot cause the changes in the permeability of hydrogels.

To further assess the applicability and reliability of the method proposed, herein, we utilize the CSDR-Sensor for the detection of cocaine in urine. Ten diluted urine samples are assayed according to general procedures, and the determined concentrations are estimated from the corresponding signals of the flow-through time and the regression equations. As shown in Fig. 6b, the flow-through time is linearly correlated with the concentration of cocaine in urine. Moreover, an adjusted $R$-square value of 0.9829 and a slope of 0.9775 are obtained. The results prove that the CSDR-Sensor has high sensitivity for the detection of cocaine in real samples and would be well suited for highly sensitive detection in clinical and POCT settings. On the other hand, DNA-hydrogel can be easily stored, which lays the foundation of the applicability of the sensor for POCTs assays. As shown in Supplementary Figure 3, after storing the sensors for a time duration (2 weeks) under a certain temperature (4 °C) in an enclosed space with atmosphere of 2 mL buffer (77 mM NaHPO$_4$, 23 mM NaH$_2$PO$_4$, 50 mM NaCl, 5 mM MgCl$_2$), the sensors shall still be able to function. With the same concentration of cocaine solution (1 µM), similar results were obtained with RSD less than 3.4%, suggesting the fabrication reproducibility and working stability of the CSDR-Sensor.

## Discussion

In comparison with conventional quantitative cocaine assays based on DNA hydrogel using colorimetric[26,34], glucose meter[35], UV–vis spectra[27], or naked eye[12,21,22] measurements, the CSDR-Sensor based on the above principles and methods shows good performance and effectiveness (Supplementary Table 2). The analysis results of our work are quantitative, and the LOD of our

method is extremely low compared with the reported methods previously. This result is comparable to the most sensitive methods based on hydrogel reported to date, and a small amount of DNA hydrogel is needed for the sensitive analysis without the aid of other instruments. Only 0.01 µL of DNA hydrogel (just a small, thin film) is required for the detection of 10 nM to 100 µM cocaine. Therefore, it is very cheap than previously reported methods. Moreover, this method is very simple for real POCT applications by just integrating the capillary action and DNA hydrogel film into the capillary tube.

In summary, a facile and sensitive visual quantitative sensor is developed by integrating the permeability change in a DNA hydrogel film with self-driven motion. The time of the sample solution flowing through a certain length of capillary tube is influenced by the different permeability of hydrogel film. The permeability has a linear relationship with the target concentration. This result is comparable to the most sensitive methods based on hydrogel reported to date, and it can be achieved by ultra-trace DNA hydrogel without the aid of other instruments. Moreover, the method developed here can be used as a powerful tool for the detection of other targets simply by replacing the linker sequences in the hydrogel. A variety of aptamers against a broad range of targets are either available or can be obtained via the systematic evolution of ligands by exponential enrichment (SELEX)[36–38]. With the advantages of high sensitivity, selectivity, and versatility, as well as the features of low cost, equipment-free, and portability, this method has great significance for developing smart hydrogel public applications.

## Methods

**Materials and reagents.** Cocaine was obtained from National Institutes for Food and Drug Control (Beijing, China). Ammonium persulfate (APS), tetra-methylethylenediamine (TEMED), and acrylamide were purchased from Sigma-Aldrich (St. Louis, MO, USA). 2-Cyanoethyl diisopropyl chlorophosphoramidite was purchased from Chem Genes (Wilmington, MA, USA). FC-40 (a mixture of perfluoro-tri-n-butylamine and perfluoro-di-n-butylmethylamine) was purchased from Minnesota Mining and Manufacturing Company (St. Paul, MN, USA). DNA synthesis reagents were purchased from Glen Research (Sterling, VA, USA). Other reagents were purchased from Sinopharm Chemical Reagent (Shanghai, China). The cocaine buffer contained 77 mM Na$_2$HPO$_4$, 23 mM NaH$_2$PO$_4$, 50 mM NaCl, 5 mM MgCl$_2$ (pH 7.3). All solutions were prepared with ultra-pure Milli-Q water (resistance > 18 MΩ cm$^{-1}$). Glass capillaries from Suzhou City Crystal Glass Co., Ltd. were used. The inner radius of these capillaries is constant and equal to 300 ± 5 µm.

**Preparation of aptamer-crosslinked hydrogel.** Synthesis of Acrylic-DMT Phosphoramidite and DNA synthesis were prepared according to the reported methods. All oligonucleotides used in the work were synthesized on a PolyGen 12-

Column DNA Synthesizer in-house according to the standard DNA synthesis protocol. Strand A and strand B were modified at the 5′-end using the acrylic phosphoramidite. The product was cleaved from the solid support, deprotection with ammonia treatment, and purified by LC3000 semi-preparative HPLC system on a reverse-phase C18 column (Chuang Xin Tong Heng, Beijing, China). A solution of 0.1 M triethylamine acetate (pH 6.5) was used as HPLC buffer A, and HPLC-grade acetonitrile (Fisher) was used as HPLC buffer B. After detritylation, the DNA were desalted with a NAP-5 column (GE Healthcare), quantified by UV–vis spectrometry, and stored at −20 °C for future use.

Stock solutions of strand A and B with the desired concentrations were prepared separately in centrifuge tubes containing 4% acrylamide. They were placed in a vacuum desiccator for 10 min at 25 °C to remove air. Then, 1.4% (v/v) of freshly prepared initiator (0.05 g of APS dissolved in 0.5 mL ultrapure water) and accelerator (25 μL TEMED dissolved in 0.5 mL ultrapure water) solution were added to both stock solutions immediately. The centrifuge tubes were placed in the vacuum desiccator again to polymerize under vacuum at 25 °C. Linear-chain PS-A and PS-B were produced after 15 min. To generate the hydrogel, 110 μM PS-A and 110 μM PS-B were mixed with a certain amount of aptamer linker in buffer (77 mM NaHPO$_4$, 23 mM NaH$_2$PO$_4$, 50 mM NaCl, 5 mM MgCl$_2$). The mixture was shaken vigorously to guarantee the homogeneity of the solution before incubating in a dry bath at 65 °C for 5 min, and then allowed to slowly cool to room temperature (repeat three times) to produce cocaine-aptamer cross-linked hydrogel.

**CSDR-Sensor design and fabrication**. In order to get more silicon hydroxyl, glass capillary tube was cleaned by piranha solution (mixture of 7:3 (v/v) 98% H$_2$SO$_4$ and 30% H$_2$O$_2$) at 100 °C for 1 h and followed by thorough ultra-sonication for 10 min in acetone and ethanol. Then the glass capillary tube was rinsed three times in Milli-Q water and blown dry with nitrogen gas. In order for the capillary to obtain the temperature which could disintegrate the DNA hydrogel of step 2, the capillary tube was heated at 90 °C for 10 min. Next, the hot capillary tubes were inserted into the DNA hydrogel by lifting machine (depth 1 mm) and taken out about 3 s later. The hydrogel film in capillary with thickness was quickly selected through a microscope. Subsequently, the capillary tube with DNA hydrogel film is fixed on a graduated plate to simply obtain CSDR-Sensor, and refrigerated storage.

**CSDR-Sensor assays**. The use of CSDR-Sensor device is as simple as using a thermometer as shown in Supplementary Movie 4. In a typical test on CSDR-Sensor, the device is placed horizontally. And the sample solution (10 nM to 100 μM cocaine solution in buffer 77 mM NaHPO$_4$, 23 mM NaH$_2$PO$_4$, 50 mM NaCl, 5 mM MgCl$_2$) is added on a carrier plate, then move the carrier to touches hydrogel film in capillary tube. Then a timer with a resolution of tens of milli-seconds was used to record the time required for the sample reagent to flow through the 8 cm-length in capillary tube.

**Rheological measurements**. Rheological measurements were carried out using a rheometer model MCR-302 (Anton Paar, Graz, Austria) equipped with a Peltier device for temperature control. During all rheological measurements, silicone oil trap was used to minimize evaporation. For characterization of DNA hydrogels, a plate–plate geometry with a 12 mm diameter upper plate was used. The upper plate was made of stainless steel and the lower plate was a Teflon Peltier surface. The upper plate was set at the desired separation distance that was equivalent to a normal force ($F_N$) of 5 N (within a linear range; the moduli are independent of the magnitude of imposed force) corrected with a "true gap" function of the instrument. The hydrogels were characterized under frequency sweep ($\omega = 0.1$–50 rad s$^{-1}$, 25 °C). The mesh size $\xi$ of the hydrogels was calculated from the storage modulus $G'$ value as follows[32]:

$$\xi = \left( \frac{RT}{G' N_A} \right)^{\frac{1}{3}} \tag{1}$$

where $R$ is the gas constant, $T$ is the measurement temperature, and $N_A$ is Avogadro's constant.

## Data availability
Data within the manuscript and its Supplementary Information are available from the corresponding author upon reasonable request. A separate source data file is provided in a persistent repository (URL: https://pan.baidu.com/s/1Wb12QMsmxVjPcZLNaWHopg; Fetch Code: dij4). The source data underlying Figs. 4a–d, 5b, c, and 6a, b and Supplementary Figures 2 and 3 are provided as a Source Data file.

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

## Acknowledgements

This work was supported by Beijing Natural Science Foundation (2172039), National Key Research and Development Program of China (2018YFD0401302), National S&T Major Project of China (2018ZX10301201), the National Science Foundation of China (21735004, 21890742, 21727815, 21521004, 21435004), and the Fundamental Research Funds for the Central Universities and USTB.

## Author contributions

Yansheng Li and Yongqiang Wen designed the experiments project. Yansheng Li carried out the measurements. Yanli Ma, Tingyu Li, Zhehao Lv, and Chaoyong James Yang prepared the DNA materials. Yongqiang Wen, Yansheng Li, and Xiangyu Jiao discussed and analyzed the data. Xueji Zhang assisted in evaluating the results. Yansheng Li and Yongqiang Wen wrote and revised the paper.

## Additional information

**Competing interests:** The authors declare no competing interests.

