## [Peer Review File · Nature Communications]

Reviewers' comments:

Reviewer #1 (Remarks to the Author):

Review for manuscript NCOMMS-18-23200

Recommendation: Major revision.

This work presents a sensor based on DNA-hydrogel; with small volume of DNA-hydrogel encapsulated in a glass capillary, this sensor transfers chemical signals (e.g. cocaine concentration, etc.) to visual signals (i.e. flow rate in glass capillary) and have the potential to lower sensor-cost and be applied for the point-of-care test (PoCT) of bioassays. The concept of this work is interesting and could have impact on the field of PoCT using DNA hydrogels. However, the manuscript contains many questionable points and the experimental design and analysis is not adequate. If the authors could improve the quality of the manuscript and append new and persuasive experimental results, I would like to provide more favorable responses.

Please find my comments in details as follow:

Major comments:

1. The manuscript failed to describe the fabrication process of the sensors in a clear and detailed fashion. The most unclear point after reading through the manuscript is: when is the step that cocaine is introduced. General readers outside of this specific field will naturally wonder between the following two possibilities: (a) Cocaine was pre-mixed with DNA-hydrogel and then introduced into glass capillary; (b) Cocaine was later introduced to the capillary that is previously filled with DNA-hydrogel. Through my knowledge and prediction, I think (b) is the correct one. However, I could not confirm whether my prediction is correct or not by searching through the manuscript. In addition, the material and method section is too short and lacks many details, please append a readable and detailed material and method section in SI.
2. Following my first comment, if (a) is the case, then the field applicability of this sensor is highly questionable. Since the preparation of this sensor shall be performed on field, which involves the heating of sample to 90 deg C.
3. Following my first comment, if (b) is correct, then the manuscript lacks analysis on the DNA-hydrogel to clarify the principal of this sensor. The authors should answer the following questions in the manuscript.
 - (1) The thickness and the effective aptamer density of the DNA-hydrogel would determine the performance of this sensor. How is the thickness of the hydrogel controlled through the preparation process? Fig S1(A) does give information on thickness determination. What is the distribution of hydrogel heights among samples?
 - (2) The effective aptamer density should be estimated through rheological analysis of the hydrogel. Does the stiffness of the DNA-hydrogel change accordingly to the addition of cocaine?
 - (3) I believe that there are two steps upon sensing cocaine with this sensor: first, cocaine reacts with DNA-hydrogel and changing hydrogel permeability; then, dyes flow through the DNA-hydrogel into capillary. The different reaction speed of these two steps are shown as the difference of traveling time for the first 0-1 cm and the rest of the length on Fig. 3B. Does the DNA-hydrogel reach stable state before dyes flow through the hydrogel? Does this first stage last for same time for all the different concentration of cocaines? Does thickness of the DNA-hydrogel change (does it swell) during the measurement? If so, does the swelling ratio change according to the cocaine concentration?
4. The authors claimed that DNA-hydrogel can be easily stored, which lays the foundation of the applicability of the sensor for PoCT assays. Please provide evidence on this point by providing reasonable test data, e.g. after storing the sensors for a time duration (e.g. 1 month) under a certain temperature and humidity, the sensors shall still be able to function. There has been too many PoCT sensors successfully demonstrated in labs, however, the storage and field-test

performances are very essential to judge the general impact of a work dedicated to PoCT assay. 5. In table 1, the authors compared this work with other previous works and claimed that other works require highly trained operator. I checked out the other works, several of them using similar bar-chart setup and can be setted just by adding solution to specific inlets, hence I cannot agree with the authors on this point.

6. The English and logic of the manuscript is rather poor. Regarding the English, I listed the questionable points (esp. wrong vocabulary usages) found only in the introduction part in my minor comments. I recommend the authors to use some English check services and have a full check of the whole manuscript. Regarding the logics, the introduction part contains twisted and cumbersome logics, especially from the 2nd paragraph of the introduction part; please try to ease the flow of logics in the introduction part.

7. The figure caption are too simple and there is even no detailed description of the figures in the main texts. For example, Figure 1 contains many symbolic icons, however, no explanation were given on them; most graphs did not indicate their N number of data points.

Minor comments:

1. Page 1, line 20: please change "in response to" to "based on".
2. Page 1, line 22: "alter" is too vague and fail to correspond "reduce" in the following text; should be either "increased" or "decreased".
3. Page 1, line 24: please change "timing" to "time duration".
4. Page 2, line 25: please change "monitor" to "quantify".
5. Page 2, line 37: vague usage of "and" in between "store" and "immobilize".
6. Page 2, line 40: there are quite a few subjective but not necessary adverbs such as "elegantly" in this line, and "cleverly" in Page 3, line 49, etc.
7. Page 2, line 45: please change "hydrogel chemical or physical changes" to "chemical or physical changes of hydrogels".
8. Page 2, line 46: cannot understand this whole line.
9. Page 3, line 52: what is morphology changes? the previous texts says the detection can be done by electrical, optical signal or volume change, but I cannot find any description on morphology. what is the difference between morphology and size?
10. Page 3, line 53: "smart" and "integrated" needs to be clearly defined.
11. Page 3, line 61: what does "this" point to?
12. Page 3, line 63: how to interpret the "quickness"?
13. Page 3, line 64: how long is "long time"?
14. Page 4, line 72: please delete "in this paper" which is repetitive considering the "Herein" in the previous sentence.
15. Page 4, line 78: "this method" should be "this sensor".
16. Page 4, line 80: "of the gel" should be moved to follow "the morphology".
17. Page 4, line 80: "This method realizes" should be "The sensor is based on"; this whole sentence is hard to understand.

Reviewer #2 (Remarks to the Author):

In this manuscript, the authors reported a facile, equipment-free and quantitative device using an ultra-trace DNA hydrogel relying on target induced capillary action change in a capillary tube. The manuscript proposed an elegant approach to change the permeability of a hydrogel film in a capillary tube, which results in the timing of the target solution flowing through a capillary tube with a specified length for visual quantitative detection. It represents an important progress in the field of DNA hydrogel-based analytical method for POCT. The manuscript is recommended for publication in the journal after the following concerns have been addressed.

1. The concept of distance-based measurement has been proposed. One paper had been reviewed

by Tian et al (Distance-based Microfluidic Quantitative Detection Methods for Point-of-Care Testing, Lab Chip, 2016, 16, 1139-1151). The authors should discuss briefly about the advantages of this method versus distance-based measurement.

2. Please discuss the limitations of the new method.

3. The "CHF sensor" in captions of figure 1 does not appear in the text. Please check and verify.

Reviewer #3 (Remarks to the Author):

The authors present a capillary-based system for the simple naked eye quantification of an analytical target, demonstrated on the example of cocaine. The working principle relies on the target-driven degradation of an aptamer cross-linked hydrogel inside a capillary, which results in modulation of liquid flow speed. The time required for the sample liquid to flow a specified distance (8 cm) corresponds to the concentration of analyte present in the sample.

Although this is a generally sound piece of work showing some impressive results in terms of detection limit (at least for the cocaine model analyte presented) and operational simplicity, I do not believe that Nature Communications is a suitable platform for the publication of this report. The main reason for this is the limited novelty over the existing literature, in particular over previously reported work by the Chaoyong Yang group. The identical analyte recognition mechanism (analyte-dependent hydrogel formation or degradation modulating flow behavior) has been used in multiple cases by that group. The current report certainly demonstrates some advantages (e.g. lower required aptamer consumption, low detection limit). But in my opinion, these are mostly related to the extension and further improvement of an already presented analytical technique, rather than a new method of high interest to the wider community. Therefore, I believe that a journal focusing on analytical chemistry or measurement technology would be a more suitable platform.

My general recommendation to the authors is to increase the level of detailedness, provide more relevant statistical information, demonstrate the application to real samples and compare with a reference method. By doing so, this work will be readily accepted by a more specialized journal in the field of analytical chemistry. Please also refer to my comments below for further comments and suggestions.

Major points regarding limited novelty

1. The novelty of this report is limited in comparison to previously published work by the Chaoyong Yang group. This applies in particular to the report in Anal. Chem. 2015, 87, 4275–4282 (ref. 12 of the current work), where the concentration of cocaine results in a visually observed change in length or change in number of colored spots on a paper device. Although the achieved LOD is far from the one reported here and the results are less quantitative, the assay can be performed within 6 min (in contrast to up to 40 min required here depending on cocaine concentration) and is user friendly.

2. The statements in Table 1 that the paper-based devices (refs. 12, 22) require a "highly-trained" operator is exaggerated. In both previous reports, the authors point out by themselves that the corresponding devices are "user- friendly" and can be operated "without trained staff". Therefore, the "user-friendliness" of the current capillary-based system compared to the reported technology is overstated in the introduction.

Specific points:

3. The current system will be significantly influenced by the viscosity of the investigated sample

liquid. This aspect is not considered in the current study and should be investigated. For example, the viscosity of urine samples significantly changes with protein or other constituent levels. This can probably be overcome by dilution (as applied here), which introduces a quantitative volumetric operation step influencing user-friendliness and error risk.

4. In my opinion, the “user-friendliness” of the current system is overrated (please refer also to point 2 above). The application of the sample (no fixed volume required) is certainly very simple. But the measurement of time as the analytical signal requires a user to permanently monitor the system (up to 40 min at low cocaine levels) in order not to miss the “assay endpoint” (liquid reaching the 8 cm marker).

5. The introduction mentions the requirement of “long time for target diffusion and reaction completion”, but as mentioned in point 4, the current system is not generally faster than the other methods mentioned in Table 1 relying on naked eye detection.

6. More statistical data on the device fabrication reproducibility is required. The hydrogel membrane at the entrance to the capillary is very thin (0.15 mm). The authors do a “quick selection through a microscope” to select suitable capillaries. What is the hydrogel thickness reproducibility? To what extent do fluctuations in thickness influence the liquid flow speed? What is the percentage of “outliers” (capillaries with hydrogels other than 0.15 mm)?

6. In the urine assay (Fig. 4B), error bars are missing.

7. How do changes in sample pH or sample ionic strength influence the liquid flow behavior through the hydrogel?

Minor points:

8. The title is hard to understand as a “self-standing” sentence. For example, the expression “self-driven motion” is used to describe simple capillary action. The expression “super sensitive” is very subjective and ambiguous.

9. Page 2, line 27: “high detection limit” should be “low detection limit”

10. The definition of SNR is not clear. Normally, a high SNR is desirable to clearly distinguish the signal from background levels. The parameter used by the authors is rather the inverse of the SNR ($1/\text{SNR}$).

11. Some relevant information should be more clearly and specifically indicated in the text or supporting information. For example, it is not clear without reading the entire text that the authors look at the flow time required to reach the 8 cm marker.

12. In Fig. S2, the reader needs to refer to the main text to identify which y-axis belongs to which data shown in the graph.

13. To what time would the concentration of 1.2 nM (LOD) correspond?

Responses to the Referees:

(Regarding to comments from reviewer 1)

Major comments:

1. The manuscript failed to describe the fabrication process of the sensors in a clear and detailed fashion. The most unclear point after reading through the manuscript is: when is the step that cocaine is introduced. General readers outside of this specific field will naturally wonder between the following two possibilities: (a) Cocaine was pre-mixed with DNA-hydrogel and then introduced into glass capillary; (b) Cocaine was later introduced to the capillary that is previously filled with DNA-hydrogel. Through my knowledge and prediction, I think (b) is the correct one. However, I could not confirm whether my prediction is correct or not by searching through the manuscript. In addition, the material and method section is too short and lacks many details, please append a readable and detailed material and method section in SI.

Reply: The authors would like to thank the referee very much for the kind evaluation and suggestions for our manuscript. As judged by the reviewer, (b) is the correct detecting step. In a typical test using CSDR-Sensor, the device was firstly placed horizontally. The sample solution was add onto a carrier plate, and then moved the carrier plate to touch the hydrogel film in capillary tube. A timer with a resolution of tens of milliseconds was used to record the flow time required for the sample through the capillary tube with certain length. According to the referee's suggestions, we have revised the corresponding descriptions including (material and method section) to make it accurate and clear, and a video was also made to describe the testing process. (Please read the blue words in the revised manuscript and video S1).

2. Following my first comment, if (a) is the case, then the field applicability of this sensor is highly questionable. Since the preparation of this sensor shall be performed on field, which involves the heating of sample to 90 deg C.

Reply: Thanks for your question. As is answered in the first question, (b) is the

correct one.

3. Following my first comment, if (b) is correct, then the manuscript lacks analysis on the DNA-hydrogel to clarify the principle of this sensor. The authors should answer the following questions in the manuscript.

(1) The thickness and the effective aptamer density of the DNA-hydrogel would determine the performance of this sensor. How is the thickness of the hydrogel controlled through the preparation process? Fig S1(A) does give information on thickness determination. What is the distribution of hydrogel heights among samples?

(2) The effective aptamer density should be estimated through rheological analysis of the hydrogel. Does the stiffness of the DNA-hydrogel change accordingly to the addition of cocaine?

(3) I believe that there are two steps upon sensing cocaine with this sensor: first, cocaine reacts with DNA-hydrogel and changing hydrogel permeability; then, dyes flow through the DNA-hydrogel into capillary. The different reaction speed of these two steps are shown as the difference of traveling time for the first 0-1 cm and the rest of the length on Fig. 3B. Does the DNA-hydrogel reach stable state before dyes flow through the hydrogel? Does this first stage last for same time for all the different concentration of cocaine? Does thickness of the DNA-hydrogel change (does it swell) during the measurement? If so, does the swelling ratio change according to the cocaine concentration?

Reply: The authors would like to thank the referee very much for the helpful suggestions for our manuscript. We have made corresponding modifications and explanations on the principle and method of this sensor. Please read the blue words in the revised manuscript.

(1) When we prepared the hydrogel film in a capillary tube, we adjusted its thickness by controlling the residence time of hot capillary tube in the pre-prepared gel. The longer the residence time in the gel, the thicker the hydrogel film in capillary tube is obtained. In the preparation process, a lifting machine was applied, and a batch of capillary sensors were produced simultaneously. Further screening was then carried

out with microscope to ensure a uniform thickness of hydrogel. In our experiments, DNA hydrogel films with a length of 0.15 ± 0.01 mm were obtained by controlling the residence time (3 seconds). Statistical analysis showed that the capillary sensors above 90% were qualified. We have made the corresponding explanation in the revised manuscript, please see the blue font section.

(2) The effective aptamer density is very important for the effectiveness of this type of sensor. In order to get the proper aptamer density, optimization experiments were carried out, where three hydrogels with different aptamer concentrations (50, 70, and 90 μ M linker DNA in the hydrogel) are prepared. From the signal-to-noise ratios (SNRs) of different DNA crosslinking densities shown in Figure S1, the sensor with cross-linked aptamer concentration of 70 μ M showed good effectiveness. Because cocaine can form complexes with the cross-linked chains in the gel, the addition of cocaine will destroy the grid structure of the gel and reduce the stiffness of gel accordingly.

(3) Thanks very much for the professional questions and helpful suggestions. We think the speculation about sensing cocaine in two steps with this sensor is reasonable.

1) According to the experimental data, the flow rate in the second stage is basically stable. Therefore, we think the DNA-hydrogel have reached stable states before dyes flow through the hydrogel.

2) Because the flow rate in the first stage is different, the first stage may last for different time for the different concentration of cocaine. The higher the concentration of cocaine is, the shorter the duration time is.

3) Based on our observation and above analysis, we think that no change or negligible change occurred in the thickness of the DNA-hydrogel during the measurement. Using this method, we broke through the dependence of DNA hydrogel qualitative change (volume change) and realized the quantitative detection by the slight microscopic changes of gel (i.e. permeability alteration) rather than macroscopic shape changes. The interaction between cocaine and the gel might cause slight changes in the internal structure of gel, while the thickness of the

DNA-hydrogel is basically constant.

4. The authors claimed that DNA-hydrogel can be easily stored, which lays the foundation of the applicability of the sensor for PoCT assays. Please provide evidence on this point by providing reasonable test data, e.g. after storing the sensors for a time duration (e.g. 1 month) under a certain temperature and humidity, the sensors shall still be able to function. There has been too many PoCT sensors successfully demonstrated in labs, however, the storage and field-test performances are very essential to judge the general impact of a work dedicated to PoCT assay.

Reply: We thank the referee very much for the kind suggestions for our manuscript. And we have supplemented the corresponding data. As shown in Figure S4, after storing the sensors under a certain condition (two weeks, 2 mL buffer in a small glass bottle (77mM NaHPO₄, 23mM NaH₂PO₄, 50mM NaCl, 5mM MgCl₂), 4□ in a refrigerator), the sensors can still be used normally. With the same concentration of cocaine solution (1 μM), similar results were obtained with RSD less than 3.4%, suggesting the fabrication reproducibility and working stability of the CSDR-Sensor.

5. In table 1, the authors compared this work with other previous works and claimed that other works require highly trained operator. I checked out the other works, several of them using similar bar-chart setup and can be setted just by adding solution to specific inlets, hence I cannot agree with the authors on this point.

Reply: Thanks very much for the referee's kind evaluation and suggestions for our manuscript. The methods mentioned in the references 12, 22 and other prior published work all require the quantification of sample volume to achieve the quantitative results, while our method does not rely on the quantification of sample volume, which reduces the difficulty compared with the previously reported work. In addition, the method we provided is also very simple in the preparation process. The hot capillaries is just needed to be inserted into the gel, and then taken out. Therefore, our method is simpler in the process of preparation and operation. Nevertheless, the claim "highly trained operator" is indeed a little overrated. And we have made some modifications

to the related descriptions according to the suggestions of reviewers, as shown in table 1.

6. The English and logic of the manuscript is rather poor. Regarding the English, I listed the questionable points (esp. wrong vocabulary usages) found only in the introduction part in my minor comments. I recommend the authors to use some English check services and have a full check of the whole manuscript. Regarding the logics, the introduction part contains twisted and cumbersome logics, especially from the 2nd paragraph of the introduction part; please try to ease the flow of logics in the introduction part.

Reply: Thanks very much for the referee's kind evaluation and suggestions for our manuscript. According to the referee's suggestion, we have further polished the language using English check service to improve the logic and readability in the revised manuscript.

7. The figure caption are too simple and there is even no detailed description of the figures in the main texts. For example, Figure 1 contains many symbolic icons, however, no explanation were given on them; most graphs did not indicate their N number of data points.

Reply: Thanks very much for the referee's kind evaluation and suggestions for our manuscript. According to the referee's suggestions, we have revised the corresponding figure captions and corresponding descriptions.

Minor comments:

1. Page 1, line 20: please change "in response to" to "based on".
2. Page 1, line 22: "alter" is too vague and fail to correspond "reduce" in the following text; should be either "increased" or "decreased".
3. Page 1, line 24: please change "timing" to "time duration".
4. Page 2, line 25: please change "monitor" to "quantify".
5. Page 2, line 37: vague usage of "and" in between "store" and "immobilize".

6. Page 2, line 40: there are quite a few subjective but not necessary adverbs such as "elegantly" in this line, and "cleverly" in Page 3, line 49, etc.
7. Page 2, line 45: please change “hydrogel chemical or physical changes” to “chemical or physical changes of hydrogels”.
8. Page 2, line 46: cannot understand this whole line.
9. Page 3, line 52: what is morphology changes? The previous texts says the detection can be done by electrical, optical signal or volume change, but I cannot find any description on morphology. What is the difference between morphology and size?
10. Page 3, line 53: “smart” and “integrated” needs to be clearly defined.
11. Page 3, line 61: what does “this” point to?
12. Page 3, line 63: how to interpret the “quickness”?
13. Page 3, line 64: how long is “long time”?
14. Page 4, line 72: please delete “in this paper” which is repetitive considering the “Herein” in the previous sentence.
15. Page 4, line 78: “this method” should be “this sensor”.
16. Page 4, line 80: “of the gel” should be moved to follow “the morphology”.
17. Page 4, line 80: “This method realizes” should be “The sensor is based on”; this whole sentence is hard to understand.

Reply: We thanks the referee very much for the suggestions about the language. We have revised the above mentioned points and further polished the unclear statements in the revised manuscript.

Reviewer #2 (Remarks to the Author):

In this manuscript, the authors reported a facile, equipment-free and quantitative device using an ultra-trace DNA hydrogel relying on target induced capillary action change in a capillary tube. The manuscript proposed an elegant approach to change

the permeability of a hydrogel film in a capillary tube, which results in the timing of the target solution flowing through a capillary tube with a specified length for visual quantitative detection. It represents an important progress in the field of DNA hydrogel-based analytical method for POCT. The manuscript is recommended for publication in the journal after the following concerns have been addressed.

1. The concept of distance-based measurement has been proposed. One paper had been reviewed by Tian et al (Distance-based Microfluidic Quantitative Detection Methods for Point-of-Care Testing, Lab Chip, 2016, 16, 1139-1151). The authors should discuss briefly about the advantages of this method versus distance-based measurement.

Reply: The authors would like to thank the referee very much for the kind evaluation and suggestions for our manuscript. Distance-based measurement is a very convenient method. Through the preparation of DNA hydrogel film in capillaries, we broke through the dependence of DNA hydrogel qualitative change (volume change) for the first time, and realized the real-time monitoring of DNA hydrogel quantitative change (gel permeability). Our method has the following advantages: 1) The preparation method is simple and innovative. Only the hot capillary tube is inserted into the gel to prepare the gel film 2). Small volume gel is used; 3) it has low detection limit. Such advantages have been further emphasize in the revised manuscript.

2. Please discuss the limitations of the new method.

Reply: DNA hydrogel was formed by the hybridization of DNA strands, which is affected by pH and ionic strength. Therefore, when testing an actual sample, we need to dilute the sample into the same buffer solution as the DNA hydrogel formation to avoid the impact of pH and ion strength on the detection system as much as possible. We have added related description in the revised manuscript.

3. The “CHF sensor” in captions of figure 1 does not appear in the text. Please check and verify.

Reply: According to the referee's suggestion, we have revised the corresponding description.

(Regarding to comments from reviewer 3)

1. The novelty of this report is limited in comparison to previously published work by the Chaoyong Yang group. This applies in particular to the report in *Anal. Chem.* 2015, 87, 4275–4282 (ref. 12 of the current work), where the concentration of cocaine results in a visually observed change in length or change in number of colored spots on a paper device. Although the achieved LOD is far from the one reported here and the results are less quantitative, the assay can be performed within 6 min (in contrast to up to 40 min required here depending on cocaine concentration) and is user friendly.

Reply: The authors would like to thank the referee very much for the kind evaluation and suggestions for our manuscript. About the limitation of this work in comparison to previously published work by the Chaoyong Yang group, we have different viewpoint.

First of all, the preparation process of gel film in our CSDR-sensor is innovative. **Based on the thermal reversible principle of DNA hydrogel and the principle of capillary action, the gel film was subtly prepared in capillary tube.** When the gel in a container was in contact with the hot capillary tube, it turned into liquid solution, which was then injected into the capillary tube by virtue of the capillary action. Subsequently, the DNA hydrogel film was formed in the capillary tube at room temperature to obtain the CSDR-Sensor. It is the first time that such principle and method was proposed to prepare hydrogel-based sensing devices and use for sensitive analysis. Also, this method has a potential to be extended to other target-responsive DNA hydrogel.

Second, the detection principle is different from other reported DNA hydrogel-based sensors. It is well known that qualitative change is caused by quantitative change, and the degradation of gel into sol is qualitative change. As mentioned by referee 3,

designing DNA hydrogel sensors using the idea of analyte-dependent hydrogel formation or degradation is really not a novel idea, but almost all those published sensors were designed using the qualitative change as we have cited shown in Table1, including Anal. Chem. 2015, 87, 4275–4282 (ref. 12 of the manuscript). As we all know, qualitative change (gel formation or degradation) requires more stimulation. However, a small number of target molecules will bind to the linker in the gel to cause slight changes in the gel (qualitative change), but will not cause the degradation of the whole gel (quantitative change). So the LOD (limit of detection) of previously published work by the Chaoyong Yang group based on gel degradation is high. **In our manuscript, we realized the quantitative detection through quantitative change (permeability alteration) of gel rather than qualitative change. By flexibly utilizing capillary tube and capillary action (self-driven motion), such slight quantitative change (permeability change) could be easily observed by the naked eye.** As far as we know, this is the first time to report DNA hydrogel sensors based on quantitative change of gel. In order to express the novelty of our work clearly and intuitively, we further revised our manuscript (please see the blue words) and made videos (video S1 and S2) to show how our work was different from the previously reported methods in Anal. Chem. 2015, 87, 4275–4282.

In addition, the analysis results of our work are quantitative, and the LOD of our method is extremely lower compared with the reported methods previously. This result is comparable to the most sensitive methods based on hydrogel reported to date, and it can be achieved by ultra-trace DNA hydrogel without the aid of other instruments. In addition, it is very cheap than the previously reported methods. Only 0.01 μL of DNA hydrogel is required for the detection of 10 nM-100 μM cocaine.

As such, we believe this work has taken an important step forward than the previous work. We have further added related description and explanation in the revised manuscript, please read the blue section.

2. The statements in Table 1 that the paper-based devices (refs. 12, 22) require a “highly-trained” operator is exaggerated. In both previous reports, the authors point

out by themselves that the corresponding devices are “user- friendly” and can be operated “without trained staff”. Therefore, the “user-friendliness” of the current capillary-based system compared to the reported technology is overstated in the introduction.

Reply: Thanks very much for the referee’s kind evaluation and suggestions for our manuscript. The methods mentioned in the references 12, 22 and other prior published work all require the quantification of sample volume to achieve the quantitative results, while our method does not rely on the quantification of sample volume, which reduces the difficulty compared with the previously reported work. In addition, the method we provided is also very simple in the preparation process. The hot capillaries is just needed to be inserted into the gel, and then taken out. Therefore, our method is simpler in the process of preparation and operation. Nevertheless, the claim “highly trained operator” is indeed a little overrated, and we have made some modifications to the related descriptions according to the suggestions of reviewers, as shown in table 1.

3. The current system will be significantly influenced by the viscosity of the investigated sample liquid. This aspect is not considered in the current study and should be investigated. For example, the viscosity of urine samples significantly changes with protein or other constituent levels. This can probably be overcome by dilution (as applied here), which introduces a quantitative volumetric operation step influencing user-friendliness and error risk.

Reply: Thanks very much for the referee’s kind evaluation and suggestions. The viscosity of the measured solution is an important parameter in the analytical process. High viscosity will limit the free motion of target molecules in a dispersed system. For samples with low viscosity, no special treatment process is required. But for special samples with high viscosity, they can be pretreated by some simple treatment methods. The addition of dilution operations has a slight increase in its operational complexity. Here, we accept suggestions of reviewers and make some modifications on related descriptions about user-friendliness in the revised manuscript.

4. In my opinion, the “user-friendliness” of the current system is overrated (please refer also to point 2 above). The application of the sample (no fixed volume required) is certainly very simple. But the measurement of time as the analytical signal requires a user to permanently monitor the system (up to 40 min at low cocaine levels) in order not to miss the “assay endpoint” (liquid reaching the 8 cm marker).

Reply: Thanks very much for the kind evaluation and suggestions. The required time is really important for analytical devices. At this point, our sensors really do not show prominent advantages. It needs about 43min to detect 10nM of cocaine, and about 3min to detect 100 μ M of cocaine by our method. The lower the concentration of the sample to be detected, the longer the time it takes. For some other POCT devices reported in the literatures, different detection time was reported. (For example, 2h in *Biosensors and Bioelectronics*, 2016, 85, 496-502; 1h in *Angew. Chem. Int. Ed.* 2014, 53, 12503-12507; and 6min in *Anal. Chem.* 2015, 87, 4275-4282). Here, we would like to accept the suggestions of the reviewer and make some modifications to the related descriptions about user-friendliness in the revised manuscript.

5. The introduction mentions the requirement of “long time for target diffusion and reaction completion”, but as mentioned in point 4, the current system is not generally faster than the other methods mentioned in Table 1 relying on naked eye detection.

Reply: Thanks very much for the kind evaluation and suggestions. We would like to accept the suggestions of reviewer and make some modifications to relevant descriptions about user-friendliness in the revised manuscript, as above mentioned in question 4.

6. More statistical data on the device fabrication reproducibility is required. The hydrogel membrane at the entrance to the capillary is very thin (0.15 mm). The authors do a “quick selection through a microscope” to select suitable capillaries. What is the hydrogel thickness reproducibility? To what extent do fluctuations in

thickness influence the liquid flow speed? What is the percentage of “outliers” (capillaries with hydrogels other than 0.15 mm)?

Reply: Thanks very much for the kind evaluation and suggestions. The gel film of 0.15 ± 0.01 mm was obtained by adjusting the time (3 seconds) of hot capillaries in the gel. We made a statistical analysis of the thickness of the gel film, and the yield of qualified product was about 90% as shown in S3. In order to ensure that the liquid velocity is not affected in our experiments, the error of the thickness of gel film we selected is defined as 0.01mm. The percentage of “outliers” (capillaries with hydrogels other than 0.15 mm) was about 10%. We have made the corresponding explanation in the revised manuscript, please see the blue font section.

7. In the urine assay (Fig. 4B), error bars are missing.

Reply: According to the referee’s suggestion, we have added error bars in Fig.4B.

8. How do changes in sample pH or sample ionic strength influence the liquid flow behavior through the hydrogel?

Reply: Thanks very much for the kind evaluation. DNA hydrogel was formed by the hybridization of DNA strands, which is affected by pH and ionic strength. Therefore, when testing an actual sample, we need to dilute the sample into the same buffer solution as the DNA hydrogel formation to avoid the impact of pH and ion strength on the detection system as much as possible. We have made corresponding explanations in the revised manuscript, please see the blue font section.

Minor points:

9. The title is hard to understand as a “self-standing” sentence. For example, the expression “self-driven motion” is used to describe simple capillary action. The expression “super sensitive” is very subjective and ambiguous.

Reply: The authors would like to thank the referee very much for the kind evaluation and suggestions for our manuscript. According to the referee's suggestion, we have revised the corresponding descriptions to make it accurate. (Please see the blue words in the revised manuscript)

10. Page 2, line 27: "high detection limit" should be "low detection limit"

Reply: The authors thank the referee for the suggestions for our manuscript. According to the referee's suggestion, we have revised the corresponding descriptions to make it accurate.

11. The definition of SNR is not clear. Normally, a high SNR is desirable to clearly distinguish the signal from background levels. The parameter used by the authors is rather the inverse of the SNR ($1/\text{SNR}$).

Reply: The authors thank the referee very much for the suggestions for our manuscript. According to the referee's suggestion, we have redefined the SNR in this paper. And we have made the corresponding explanation in the revised manuscript, please see the blue font section.

12. Some relevant information should be more clearly and specifically indicated in the text or supporting information. For example, it is not clear without reading the entire text that the authors look at the flow time required to reach the 8 cm marker.

Reply: According to the referee's suggestion, we have revised the corresponding descriptions to make it accurate and clear. (Please see the blue words in the revised manuscript)

13. In Fig. S2, the reader needs to refer to the main text to identify which y-axis belongs to which data shown in the graph.

Reply: According to the referee's suggestion, we have revised the corresponding descriptions to make it accurate. (Please see the blue words in the revised manuscript in Figure S2).

14. To what time would (LOD) correspond?

Reply: Thanks for the referee's the kind evaluation for our manuscript. The lower the concentration of the sample to be detected, the longer the time it takes. The time is about 50 min for the concentration of 1.2 nM.

Reviewers' comments:

Reviewer #1 (Remarks to the Author):

Review for manuscript NCOMMS-18-23200

Recommendation: Rejection.

Through this version of revision, the authors have addressed few of my previous major comments (#1, #2, #4, #6, #7). However, there are still two major concerns not resolved. First, regarding my previous comment (Major, #3), the author still did not provide feasible material characterization such as rheological analysis of the hydrogel. In addition, the optimization of the DNA-aptamer is not fully convincing and the theory relative to the optimization needs to be established and verified; since the theory lays the foundation for the further development of such kind of sensors, it shall be an essential part of this publication. Second, regarding my previous comment (Major, #5), I am still not convinced that the sensor scheme provided here is a huge leap forward in this field by the author's comparisons.

As minor concerns, the manuscript and relative materials still have many errors such as the floating mark on the left bottom of the supplementary video s1 and s2 (the last few frames in these videos are also unnecessary).

Reviewer #2 (Remarks to the Author):

I looked into the revised manuscript and agreed that the authors have responded to all my comments satisfactorily. Thanks.

Reviewer #3 (Remarks to the Author):

As already stated in my comments for the originally submitted version of this manuscript, I regard this contribution as being scientifically sound and showing impressive results. My original concerns were related to some limitations in novelty in light of the authors' previously published work.

In their revision and in their comments to my previous questions, the authors have managed to more clearly point out the differences and advances made over their previous work. In addition, they have also satisfactorily answered most of my specific questions. Finally, the revised manuscript does no longer include some statements that I have regarded as somewhat overrated. For these reasons, I can recommend publication of this work.

There is just one point that the authors might want to consider revising:

The sentence on page 2 of the revised manuscript reading " It is well known that qualitative change is caused by quantitative change. The degradation of gel into sol is qualitative change, which is the result of quantitative change to a certain extent" is very hard to understand. The use of the expressions "qualitative" and "quantitative" change is probably not very appropriate to distinguish between situations where the permeability of the gel is changed or where the gel is completely decomposed. I recommend to consider a re-wording of this section.

Responses to the Referees:

(Regarding to comments from reviewer 1)

Reviewer #1 (Remarks to the Author):

1. Through this version of revision, the authors have addressed few of my previous major comments (#1, #2, #4, #6, #7). However, there are still two major concerns not resolved. **First**, regarding my previous comment (Major, #3), the author still did not provide feasible material characterization such as rheological analysis of the hydrogel. In addition, the optimization of the DNA-aptamer is not fully convincing and the theory relative to the optimization needs to be established and verified; since the theory lays the foundation for the further development of such kind of sensors, it shall be an essential part of this publication.

Reply: The authors is very grateful to the referee for the suggestions on our manuscript before and this time.

We have further considered the suggestions, supplemented the relevant experiments and further revised the paper. We believe that these changes have further improved the manuscript, and we hope that the current version would meet high standards for *Nature Communications*.

As for the first concern of the reviewer, first of all, the rheological analysis of the hydrogel has been supplemented in our revised manuscript, please see Figure 3B, C and D. The linear viscoelastic interval of hydrogels and the temperature responsive property of DNA hydrogels were obtained by rheological analysis. In order to get the proper aptamer density for the detection of cocaine in a certain concentration range, optimization experiments were carried out. Three well-formed gels prepared from different concentrations of aptamer were first selected for further optimization. From the signal-to-noise ratios (SNRs) of different DNA crosslinking densities shown in Figure 3A, the sensor with 70 μM cross-linked aptamer concentration showed good effectiveness. Furthermore, the storage modulus G' value of DNA hydrogel with different concentration DNA linker were characterized using rheology (Figure 3D). As can be seen from Figure 3D, in the frequency sweep, the storage modulus G' value of hydrogel increased from 10 to 900 Pa with the increase of DNA linker concentration, which can be attributed to the densification of the hydrogel network (*Macromolecules* 1987, **20**, 2226-2237. *Electrophoresis* 2006, 27, 3349-3358). Considering the result of Figure 3A, the capillary flow velocity of the sample gradually decreases with the increase of aptamer

concentration, which is consistent with the storage modulus analysis results. The larger the storage modulus is, the smaller the hydrogel mesh is, and the corresponding permeability becomes lower. The relationship among aptamer concentration, gel permeability and the flow time of samples confirmed the feasibility of controlling capillary flow rate by regulating the permeability of gel membrane, which laid the foundation for our experiments. Here, we would like to accept the suggestions of the reviewer and make some modifications to the related descriptions in the revised manuscript, please read the blue section.

2. Second, regarding my previous comment (Major, #5), I am still not convinced that the sensor scheme provided here is a huge leap forward in this field by the author's comparisons.

Reply: As for the second concern of the reviewer about innovation and comparison, we have further summarized the characteristic and revised the article, and we hope that this will help the reviewer and readers to better understand our work. Firstly, the detection principle is different from other reported DNA hydrogel-based sensors. It is well known that any qualitative change in performance comes from the accumulation of quantitative change. As concerned by the referee, designing DNA hydrogel sensors are really not novel, but almost all those published sensors were designed based on the analyte-dependent hydrogel formation or complete degradation (i.e. qualitative transformation between gel and sol), which usually achieved qualitative detection or need other auxiliary means to help characterize the qualitative change to achieve quantitative detection. In addition, those studies only focused on macro-morphological changes, without considering micro-changes and processes. Furthermore, such qualitative transformation (gel formation or degradation) requires more stimulation. Therefore, a large number of target molecules are used, and the LOD (limit of detection) in previously published work based on gel complete degradation is relatively high. However, a small number of target molecules will bind to the linker in the gel to cause slight changes in the gel (quantitative change), but will not cause the degradation of the whole gel (qualitative change). **In our manuscript, we realized the quantitative detection through quantitative change (permeability alteration) of gel rather than qualitative change (complete gel degradation). By flexibly utilizing capillary tube and capillary action (self-driven motion), such target-induced slight quantitative change (permeability change) could be directly observed by the naked eye, and no other auxiliary means are needed.** As far as we know, this is the first time to report such DNA hydrogel sensors based on quantitative change of gel.

Secondly, the preparation method of gel film in our CSDR-sensor is innovative. In almost all those reported DNA hydrogel-based sensors, in order to facilitate the observation and utilization of sol-gel transformation, gels are usually prepared in large containers, where a large volume of gel was used and no special preparation methods were used. Even with the use of microfluidic channel in the experiment, the purpose is to convert the qualitative transformation between gel and sol in large containers into visible readings. Compared with those published sensors, in our experiment the gel film was directly prepared in capillary tube based on the thermal reversible principle of DNA hydrogel and the principle of capillary action. When the gel in a container was in contact with the hot capillary tube, it turned into liquid solution, which was then injected into the capillary tube by virtue of the capillary action. Subsequently, the DNA hydrogel film was formed in the capillary tube at room temperature to obtain the CSDR-Sensor. It is the first time that such principle and method was proposed to prepare hydrogel-based sensing devices and use for sensitive analysis.

In addition, the CSDR-Sensor based on the above principles and methods show good performance and effectiveness. The analysis results are quantitative, and the LOD of our method is extremely lower compared with the reported methods previously. This result is comparable to the most sensitive methods based on hydrogel reported to date, and it can be achieved by ultra-trace DNA hydrogel (just a small, thin film) without the aid of other instruments. Only 0.01 μL of DNA hydrogel is required for the detection of 10 nM-100 μM cocaine. Given that DNA synthesis is relatively expensive, our method is very cheap than the previously reported methods. Because of the design flexibility of DNA and aptamer, such sensor could be expected to find more practical and wide applications.

As such, we believe this work has taken an important step forward than the previous work and Nature Communication is the right journal for this research. In order to express the novelty of our work clearly and intuitively, we further revised our manuscript (please see the blue words). We would like to thank the referee again for the helpful suggestions. We have further revised related description and explanation in the revised manuscript to express more accurately. Please reconsider our revised manuscript.

3. As minor concerns, the manuscript and relative materials still have many errors such as the floating mark on the left bottom of the supplementary video s1 and s2 (the last few frames in these videos are also unnecessary).

Reply: Thanks very much for the referee's helpful suggestions. According to the referee's suggestions, we have revised the above mentioned video and further polished the unclear statements in the revised manuscript.

(Regarding to comments from reviewer 2)

Reviewer #2 (Remarks to the Author):

I looked into the revised manuscript and agreed that the authors have responded to all my comments satisfactorily. Thanks.

Reply: Thanks very much for the referee's positive evaluation on our manuscript.

(Regarding to comments from reviewer 3)

Reviewer #3 (Remarks to the Author):

As already stated in my comments for the originally submitted version of this manuscript, I regard this contribution as being scientifically sound and showing impressive results. My original concerns were related to some limitations in novelty in light of the authors' previously published work.

In their revision and in their comments to my previous questions, the authors have managed to more clearly point out the differences and advances made over their previous work. In addition, they have also satisfactorily answered most of my specific questions. Finally, the revised manuscript does no longer include some statements that I have regarded as somewhat overrated. For these reasons, I can recommend publication of this work.

There is just one point that the authors might want to consider revising:

The sentence on page 2 of the revised manuscript reading " It is well known that qualitative change is caused by quantitative change. The degradation of gel into sol is qualitative change, which is the result of quantitative change to a certain extent" is very hard to understand. The use of the expressions "qualitative" and "quantitative" change is probably not very appropriate to distinguish between situations where the permeability of the gel is changed or where the gel is completely decomposed. I recommend to consider a re-wording of this section.

Reply: Thanks very much for the referee's kind evaluation and suggestions for our manuscript. According to the referee's suggestions, we have revised the corresponding descriptions, and we also have reduced the use of "qualitative" and "quantitative" and replaced them directly with "permeability change" and "complete degradation" to make it accurate and easy to understand.

REVIEWERS' COMMENTS:

Reviewer #1 (Remarks to the Author):

Review for manuscript NCOMMS-18-23200

Recommendation: Minor Revision.

Through this version of revision, the authors have addressed most of my previous major comments. Regarding my previous comment (Major #3), the authors provided material characterization such as rheological analysis of the hydrogel. In addition, the optimization of the DNA-aptamer is now convincing and the theory relative to the optimization was established and verified. Second, regarding my previous comment (Major, #5), I am convinced that the sensor scheme provided here is innovative and might have broad impact to this field.

Before recommending publication, some minor concerns shall be resolved.

First, some graphs did not indicate the N numbers, e.g. Figure 3(A-D), the authors shall check out all the figures and add N numbers either in the texts or captions.

Second, some figure numbers seem to be wrong, e.g. in first paragraph of section 2.5, all the "Figure 3*" seems to be pointing to the actual "Figure 4*", the authors shall check out all such citations to the figures.

Third, I would recommend Table 1 to be moved to supplementary materials.

Fourth, I agree with the other reviewers that the authors' usage of "qualitative" and "quantitative" is very hard to follow, e.g. in second paragraph of section 1, the "which can usually achieve qualitative detection or need other auxiliary means to help characterize the qualitative change to achieve quantitative detection" sentence is too long and too hard to understand, the authors shall check out all their sentences which used the "qualitative/quantitative".

Fifth, I would recommend professional English checking services for all the final manuscript if possible, since there is still apparent grammar errors, e.g. in the first reply of the rebuttal, the sentence "The authors is very grateful to the referee..." shall apparently be "The authors are very grateful to the referee".

Finally, regarding Scheme 1, despite the fact that the graphics are simple to read and the iconic language (using "soldiers/tanks") does convey some correct informations, the scheme lacks detailed description of the scientific ramifications. In other words, the current Scheme 1 is rather suitable to be used as "graphic abstract", than to be used as conceptual scheme in the main manuscript. For professional readers, they would expect more scientific details rather than the simple metaphoric icons.

Responses to the Referee:

Reviewer #1 (Remarks to the Author):

Review for manuscript NCOMMS-18-23200

Recommendation: Minor Revision.

Through this version of revision, the authors have addressed most of my previous major comments. Regarding my previous comment (Major #3), the authors provided material characterization such as rheological analysis of the hydrogel. In addition, the optimization of the DNA-aptamer is now convincing and the theory relative to the optimization was established and verified. Second, regarding my previous comment (Major, #5), I am convinced that the sensor scheme provided here is innovative and might have broad impact to this field.

Before recommending publication, some minor concerns shall be resolved.

1. First, some graphs did not indicate the N numbers, e.g. Figure 3(A-D), the authors shall check out all the figures and add N numbers either in the texts or captions.

Reply: Thanks very much for the referee's helpful suggestions on our manuscript. We have checked out all the figures and make corresponding modifications.

2. Second, some figure numbers seem to be wrong, e.g. in first paragraph of section 2.5, all the "Figure 3*" seems to be pointing to the actual "Figure 4*" , the authors shall check out all such citations to the figures.

Reply: Thanks very much for the referee's careful review and helpful suggestions. According to the referee's suggestions, we have revised this error in the revised manuscript.

3. Third, I would recommend Table 1 to be moved to supplementary materials.

Reply: The authors is very grateful to the referee for the helpful suggestions on our manuscript. We have move Table 1 to supplementary materials in the revised manuscript.

4. Fourth, I agree with the other reviewers that the authors' usage of "qualitative" and "quantitative" is very hard to follow, e.g. in second paragraph of section 1, the "which can usually achieve qualitative detection or need other auxiliary means to help characterize the qualitative change to achieve quantitative detection" sentence is too long and to hard to understand, the authors shall check out all their sentences which used the "qualitative/quantitative" .

Reply: Thanks very much for the referee's kind suggestions for our manuscript. According to the referee's suggestions, we have further revised the corresponding descriptions and reduced the use of "qualitative" and "quantitative" to make it accurate and easy to understand.

5. Fifth, I would recommend professional English checking services for all the final manuscript if possible, since there is still apparent grammar errors, e.g. in the first reply of the rebuttal, the sentence "The authors is very grateful to the referee..." shall apparently be "The authors are very grateful to the referee" .

Reply: We thanks the referee very much for the suggestions about the language. We have further polished the language to express more accurately in the revised manuscript.

6. Finally, regarding Scheme 1, despite the fact that the graphics are simple to read and the iconic language (using "soldiers/tanks") does convey some correct informations, the scheme lacks detailed description of the scientific ramifications. In other words, the current Scheme 1 is rather suitable to be used as "graphic abstract" , than to be used as conceptual scheme in the main manuscript. For professional readers, they would expect more scientific details rather than the simple metaphoric icons.

Reply: We thanks the referee very much for the helpful suggestions. We have added some related description in the illustration of Scheme 1 in the revised manuscript.